# Controlled Trial Examining the Strength-Based Grit Wellbeing and Self-Regulation Program for Young People in Residential Settings for Substance Use

**DOI:** 10.3390/ijerph192113835

**Published:** 2022-10-24

**Authors:** Catherine A. Quinn, Zoe C. Walter, Dominique de Andrade, Genevieve Dingle, Catherine Haslam, Leanne Hides

**Affiliations:** 1National Centre for Youth Substance Use Research, The University of Queensland, Brisbane 4072, Australia; 2School of Psychology, The University of Queensland, Brisbane 4072, Australia; 3School of Psychology, Deakin University, Melbourne 3125, Australia

**Keywords:** substance abuse, drugs, wellbeing, self-regulation, treatment, youth

## Abstract

This cohort-controlled trial examined whether the 12-session Grit Wellbeing and Self-regulation Program enhanced the treatment outcomes of young people accessing residential alcohol and other drug (AOD) treatment. Grit focuses on increasing wellbeing and reducing substance use and mental health problems by building self-regulation skills, strengths, social connections, and health behaviours. Participants were 194 (66% male, Mage 27.40) young people (aged 18–35 years) accessing a six-week residential treatment program for substance use. Participants received standard treatment, or standard treatment plus Grit (two sessions/week for six weeks). The primary outcome was substance use, measured as: (i) global substance use and (ii) alcohol, methamphetamine, and cannabis use involvement. Secondary outcomes included wellbeing, depression, anxiety, and vocational engagement. Participants were assessed at baseline, and 6-weeks (secondary outcomes only), 3-months, 6-months, and 12-months post-program enrolment. Results revealed that both groups showed a significant improvement in all outcomes at three months, and improvements were maintained at 6- and 12-month follow-ups. The Grit group had a larger reduction in methamphetamine and cannabis use involvement compared to the control group. This study presents promising evidence that a six-week residential program can achieve improvements in AOD use, depression, anxiety, wellbeing and vocational engagement. Further, targeting self-regulation may enhance such programs.

## 1. Introduction

Treatment within residential alcohol and other drugs (AOD) services is commonly offered to help-seeking individuals with moderate to severe substance use disorders (SUD) [1]. Despite some studies reporting the positive effects of residential AOD treatment, quality evidence of its efficacy is limited; particularly in young people [1,2]. There is a clear need to continue refining AOD treatment in residential settings to improve long-term outcomes. While many residential treatment programs focus on AOD use and mental health problems to some extent, they often lack a broader focus on social, emotional, physical, and psychological wellbeing. A recent systematic review on the effectiveness of residential treatment services for individuals with substance use disorders highlighted that current evidence suggest best practice approaches is residential treatment that integrates mental health treatment takes a holistic approach to improving the overall wellbeing of the individual (beyond substance dependence). This approach is also congruent with the World Health Organisation’s definition of health, which notes that health is the state of physical, mental, emotional, and social wellbeing, rather than just an absence of disease [2]. Delivering adjunctive treatments that address the complex health and wellbeing needs of people with SUDs accessing residential AOD treatment settings could improve outcomes [3]. For example, several randomized controlled trials (RCTs) have demonstrated the benefits of delivering mindfulness-based relapse prevention (MBRP) group programs in residential settings [4,5,6]. Witkiewitz and colleagues [4] found females convicted of criminal offences (*n* = 115) who received up to 8 sessions of adjunctive MBRP reported significantly fewer days of AOD use and legal problems at 15 weeks post-treatment than those who received standard residential care alone. Similarly, Davis et al. [6] found young adults (*n* = 79) who received 8 sessions of MBRP reported lower levels of AOD use and craving at 28 weeks post treatment. However, Shorey et al. [7] found no differences in craving, psychological flexibility, or dispositional mindfulness outcomes at 4 weeks post treatment, in an RCT comparing 8 biweekly sessions of group MBRP/acceptance and commitment therapy (ACT)/Mindfulness with standard residential care. In addition, interventions targeting healthy lifestyles and behaviours have also been recommended as adjunctive interventions in residential settings, with growing evidence of promising results [8,9]. For instance, Kelly et al. [8], found the delivery of up to 8 sessions of a healthy recovery group targeting smoking, diet and physical inactivity, among current smokers in residential care settings, resulted in better smoking and diet outcomes at 2 and 8 months follow up in a cluster RCT.

While these adjunctive groups programs have shown promising results within residential care settings, they only directly target some aspects of wellbeing. The use of more integrated and holistic programs that simultaneously target AOD use, mental ill-health, and the multidimensional aspects of wellbeing, could improve the outcomes of residential treatment. There is growing evidence for transdiagnostic treatments that cut across diagnostic boundaries to target the risk and protective factors which underlie multiple mental health problems. Theoretically, transdiagnostic approaches suggest that there are common latent factors that underlie a range of mental health and behavioural presentations, including SUD [10,11]. Transdiagnostic factors related to substance use include impulsivity, sensation seeking, coping, self-control, and emotion regulation skills [12,13,14,15,16,17,18]. Evidence-based treatments that have addressed these transdiagnostic factors include transdiagnostic CBT [10,19,20,21], mindfulness-based [22] and emotion-regulation focused interventions [14] for depression, anxiety and/or SUDs. However, limited research has combined interventions targeting the common mechanisms underlying AOD problems and mental ill-health, as well as the multidimensional components of wellbeing [2]. 

The Grit Wellbeing and Self-regulation Program draws upon evidence-based treatments to specifically target all four components of wellbeing: emotional, social, psychological, and physical [2,23], while simultaneously targeting the core mechanisms underlying primary SUDs and comorbid mental health problems (e.g., self-regulation, affect regulation, impulsivity; [10,11,14]). A feature of the program is that it takes a strength-based approach, moving the focus away from the deficit or pathologised approach to substance use and focuses on the strengths and resources of the clients. Adopting a strength-based approach is congruent with the program aims of holistically targeting an individual’s wellbeing, and there is emerging evidence that the utilisation of a strength-based approach is effective for supporting an individual’s substance use recovery [24,25]. The program was originally developed for, and trialled with, disengaged adolescents attending vocational schools [26], before being adapted for young people attending residential treatment facilities for substance use. 

The aim of this cohort-controlled trial was to determine whether the addition of the Grit Wellbeing and Self-regulation Program to standard residential AOD treatment would result in greater reductions in the primary outcome of substance use and secondary outcomes of wellbeing, mental health, and vocational engagement at follow-ups than standard care alone. It was expected that Grit, in combination with standard residential AOD treatment, would (i) reduce global AOD scores, (ii) lower scores on primary presenting drugs of concern (i.e., alcohol, methamphetamine and cannabis) substance involvement scores, and (iii) enhance wellbeing and vocational engagement, and reduce mental ill-health compared to standard residential AOD treatment alone at 3 months post-treatment. 

## 2. Materials and Methods

### 2.1. Study Design

This controlled trial was conducted in two residential AOD facilities run by the same organisation between 2019–2020. Both facilities delivered the same standard six-week treatment program, but one ran Grit in addition to standard care (intervention group) while the other functioned as a control. It was not possible to randomise participants between the residential facilities, as clients are admitted to the treatment facility based on its proximity to their home. Randomisation within the residential facilities was also not possible due to the potential contamination effects, as clients live together in close proximity. In this context, a quasi-experimental controlled cohort approach was considered the most feasible approach. A comparison of service file notes, conducted in 2017, revealed that clients’ in our target population of 18–35 year olds had similar demographic characteristics, primary drugs of concern, and retention and readmission rates between the two locations. This trial was registered (ACTRN12617001451392), and follows SPIRIT (Standard Protocol Item Recommendations for Intervention Trials; Ref. [16] and CONSORT (Consolidated Standards of Registering Trials) guidelines [27]. 

### 2.2. Participants

#### 2.2.1. Setting

The intervention site had 42 beds and provided treatment to young people aged 18–35. The control site was a 37 bed facility for adults, with a typical age range between 18–65 years. Standard processes were followed for client allocation to residential site. 

#### 2.2.2. Inclusion Criteria

Participants were required to be aged between 18 and 35 years and be a current client of one of the residential AOD treatment facilities. They were excluded if the treatment team assessed them to have insufficient English or severe mental illness or intellectual disability that precluded their ability to provide informed consent. At the beginning of the trial, participants were also excluded if they were returning from recent admission into a residential facility (within the past 3 weeks).

#### 2.2.3. Recruitment and Consent

All individuals who were admitted to the two residential AOD treatment facilities and met eligibility criteria were informed about the study by the intake staff. Consent was sought to participate in the evaluation, before completing routine outcome measures. All clients were assured that they would receive the same treatment opportunities irrespective of taking part in the evaluation. Ethics approval to conduct the study was obtained from the relevant university committee (approval number #2017001524).

#### 2.2.4. Sample Size

A power analysis was conducted using GLIMMPSE (http://glimmpse.samplesizeshop.org/, accessed on 10 October 2017). Based on a mixed effect repeated measures model (MMRM), power of 0.9, Type I error rate of 0.05, 2 (treatment condition) to 1 (control condition) allocation (due to disproportionate numbers of young people attending the two facilities), and previously published mean and standard deviation estimates for ASSIST scores [28], 141 participants were required to achieve moderate treatment effects. With an estimated 30% attrition rate [3], the ideal recruitment sample was estimated as 202 participants (135 in treatment condition; 67 in control condition).

### 2.3. Intervention

#### 2.3.1. Six Week Residential Treatment Program

Both residential facilities offer a treatment program delivered within six-week cycles. The program includes regular group sessions covering substance use (e.g., triggers, cravings) and emotion management, one on one counselling/case management, as well as creativity, exercise, nutrition, and relaxation activities. Clients take part in activities and regular chores that provide practical life skills for independent living. Appointments with health professionals (e.g., general practitioner, psychologist, psychiatrist) are also available, and clients are able to attend off-site 12 step fellowship meetings. In addition to the program, pre/post treatment teams assist with admission into the residential facility and the transition back to the community. Transitional accommodation and community-based counselling support is also available. Regular staff at the residential facilities include a team leader, case managers/treatment facilitators, support workers, and a registered nurse.

#### 2.3.2. Grit Program

Grit is a strengths-based, wellbeing and self-regulation program consisting of 12 (60-min) sessions, with two sessions delivered one day every week. This open group program was designed to suit the rolling intake model of the residential programs, with key information on the core Grit skills reinforced at the beginning of each session for the new participants. There was a heavy focus on teaching skills using experiential learning processes. A comprehensive summary of the program is provided in Appendix A.

Session 1 and 2 focused on grounding and mindfulness techniques, based on Rock and Water [29,30] and mindfulness-based relapse prevention programs [31,32]. Session 3 and 5 focused on social identity, relationships, and supports, and used social identity mapping activity, which has been used in numerous other programs [33,34,35,36,37,38,39]. Session 4 drew from the strengths-based mindfulness approach [25], focusing on the recognition and development of personal character strengths. Session 6 helped participants evaluate important aspects of their self-concept and identity and how their AOD use, character strengths, and social connections align [40,41]. Session 7 focused on communication and boundaries. In sessions 8 and 9, participants learnt how to identify and experience a range of pleasant and unpleasant emotions using music as a central medium. These sessions drew on principles of the Tuned In music emotion regulation program [42,43] and the “Music eScape” app [44]. Session 10 focused on the management of thoughts and cravings. Session 11 focused on healthy sleep, diet, and physical activity, and the impact of AOD use on these health-related behaviours. In session 12, participants focused on relapse prevention and recovery goals, identified instances of potential vulnerability, and developed practical strategies to manage conflict and difficult situations [45]. Participants received handouts to remind them of key exercises from each session, and developed goals at the end of each session for the following week. When a client had completed 80% of 12 sessions, they received a certificate.

#### 2.3.3. Grit Training, Facilitation, and Fidelity

Grit was co-facilitated by a clinical researcher with post graduate training in clinical psychology, and an AOD worker from the residential facility. All Grit facilitators attended a two day-training workshop, were provided with a comprehensive manual of the program, and had supervision weekly (for clinical researchers) or fortnightly (for AOD workers). Due to the importance of maintaining client confidentiality and limitations in the availability of visual and audio recording equipment, sessions could not be recorded to check fidelity of group content. Session checklists were completed at the conclusion of each Grit session, which were reviewed in supervision to evaluate delivery of content, and the treatment received by the clients. Grit attendance was monitored weekly by the treatment facilitator. Reasons for non-attendance were recorded. 

### 2.4. Measures

#### 2.4.1. Demographic and Control Measures

Demographic information included age, gender, country of birth, ethnicity, postcode, years of education completed, relationship status, whether the participants have children, employment status, living arrangements and past month homelessness, justice experiences (arrested or incarcerated) and hospitalization. The 5-item Primary Care Post-Traumatic Stress Disorder screen [46,47] and the 7-item psychosis screen [48] were also included as potential control measures. The number of days in residential treatment, and transitional housing, as well as the reason for discharge was also recorded. Information about access to other treatment and support services during residential treatment and follow-up was also collected. 

#### 2.4.2. Key Primary Outcomes

Two substance use primary outcomes were examined: (i) global substance use scores and (ii) substance involvement score on the most common primary presenting drugs of concern at the residential AOD treatment services (i.e., alcohol, methamphetamine and cannabis). Substance use was measured by the World Health Organisation Alcohol, Smoking and Substance Involvement Screening Test (WHO ASSIST), which uses 7-items to assess the frequency and related-problems of 10 classes of substance use (tobacco, alcohol, cannabis, cocaine, amphetamine, inhalants, sedatives, hallucinogens, opioids, other) over the previous 3 months, with an additional 8th item to measure injection of any drug. The reliability and validity of the ASSIST have been demonstrated across different age and cultural groups [28,49,50]. The ASSIST global substance use score has a range of 0 to 456 for all substances. The substance involvement score for the main primary presenting drugs of concern ranges from 0 to 39. The reliability for the specific scores was acceptable for all timepoints (Cronbach’s α = 0.73 to 0.89). Given the high variability in the types of substances used, the global reliability score could not be calculated. This measure was collected at baseline and 3, 6, and 12 month follow-ups.

#### 2.4.3. Other Outcomes

Wellbeing was assessed using the 14-item Mental Health Continuum—Short Form (MHC-SF), which asks participants about their emotional, psychological, and social wellbeing in the past month [23]. Each item was measured on a 5-point Likert scale (0 = never to 5 = everyday). The MHC-SF has been found to be reliable and valid in young adults [51]. A total score was computed which ranged from 0 to 70, with high reliability of scores across time (α = 0.95 to 0.96). 

Past month vocational engagement was indexed by using a composite score from two items on the Australian Treatment Outcome Profile (ATOP) [52]. Individuals would score ‘1’ if they had engaged in work or study in the past four weeks, and ‘0’ if they had not engaged in any work or study. 

Mental health was assessed using the 9-item Patient Health Questionnaire (PHQ-9) to index depression [53] to measure past 2 week depression and the Generalised Anxiety Disorder 7-item Scale (GAD-7) [54] to measure past 2 week anxiety. Both measures are scored on a 4-point scale (0 = not at all to 3 = nearly every day). 

### 2.5. Procedure

Participants completed online surveys on five occasions: baseline (upon entry to the residential treatment facility), at the end of the six-week program, and at 3-, 6-, and 12-months post-baseline. Participants were sent SMS and email reminders when surveys were due, and non-completers were contacted via phone to complete the survey with a researcher blind to treatment condition. Participants were reimbursed $20 for each follow-up survey they completed (maximum $80).

### 2.6. Data Analysis

Preliminary logistic regressions were conducted to check for baseline group differences on demographic, primary drug, and mental health factors (i.e., PTSD screen, psychosis screen, depression and anxiety). A similar analysis was conducted to detect differences between those with missing data and those without. We report below any significant differences which were controlled in subsequent analyses to address sampling bias and completion bias. The data were also checked for extreme observations (univariate Z score > ±3.29 and/or extreme outliers ± 3 SD from the treatment group mean) for all primary outcome variables. Additional logistic analyses were conducted to examine key treatment factors that may have impacted on treatment outcomes (i.e., treatment completion, re-admissions and reasons for discharge). Grit attendance over time was also examined.

To determine whether there were group differences over time on outcomes, a series of mixed effects model repeated measures (MMRM) analyses were performed [55,56,57]. Outcomes were assessed comparing baseline and 3-, 6-, and 12-month follow-ups for substance use and vocational engagement, and comparing baseline to post-treatment (6 weeks) and 3-, 6-, and 12-month follow-ups for wellbeing and mental health. This method enables an intention-to-treat approach to be adopted, with all baseline data included for those who were eligible and consented to take part in the trial (*n* = 194), even if they did not complete all follow-up surveys. All follow-up data were included, regardless of whether the participants completed either Grit, or their six weeks of residential treatment. The mixed method approach included time (baseline, 6-weeks, 3-, 6-, 12-months), group (intervention: standard treatment + Grit, control: standard treatment), and a time x group interaction as fixed effects. Additional control variables identified in the preliminary analyses were also included as fixed effects (age and having a child). Participant ID was included as a random effect. 

The Statistical Package for the Social Sciences (SPSS, version 25) was used for all analyses. Significant effects were probed using estimated marginal means comparisons, and Cohen’s d effect sizes were calculated using standard deviations pooled across groups and time.

## 3. Results

### 3.1. Participants

A total of 230, 18–35 year olds entered the residential facilities during the 18-month trial period. Clients deemed ineligible had recently attended the residential facility at the beginning of the trial. For the Grit and control sites, 84% (*n* = 118) and 85% (*n* = 76), respectively, consented to participate in the study (see Figure 1). Survey completion rates at follow-up were 81% (*n* = 157) at 6-weeks, 74% (*n* = 144) at 3-month, 64% (*n* = 120) at 6-month, and 61% (*n* = 120) at 12-month (see Figure 1). 

Baseline sample characteristics are provided in Table 1. The main drugs of concern at the time of admission were 45% amphetamines (*n* = 87, 93% ice/crystal methamphetamine), 34% alcohol (*n* = 67) and 15% cannabis (*n* = 29). There was no difference between groups in primary drug type of concern (see Table 1). Participants had used on average 6.84 (SD = 2.87) substances (from a total of 11 categories) in their lifetime, with 82% scoring in the high risk range (27+) for at least one substance (71% for alcohol and/or methamphetamine), and 99% scoring in the moderate range (4+) for at least one substance. 

On average participants scored in the high-risk range for 1.88 (SD = 1.81) substances, and the moderate range for 4.50 (SD = 2.47) substances. Participants in the control group were more likely to be older and to have a child compared to the intervention group. There was no difference between the groups on Primary Drug of concern. The intervention group were more likely to have used cannabis than the control group (80.5% vs. 65.8%, *p* = 0.023). 

There was no significant difference between the groups on the number of follow-up surveys participants completed. Eight outliers (3 SD above or below mean) were detected for the global ASSIST score (two at Baseline, two at 12-weeks, two at 6-months and four at 12-months), one for the MHC (at 6 weeks). Analyses were conducted with these outliers included and removed, with no difference in findings, so the outliers were included in the analysis for intent to treat purposes.

### 3.2. Treatment

There was no difference between groups in the average number of days in residential treatment at follow-up time-points (see Table 2). The average number of days in residential facilities for the total sample was 29.05 days (SD = 13.66) at 6 weeks, 37.71 days (SD = 22.58) at 3-months, and 43.90 days (SD = 31.89) at 12-months. Grit participants were more likely to be readmitted following discharge than the control group participants, however, there was no significant difference between groups in reasons for discharge. New readmission episodes did not differ between the groups at 3-months or 12-months. Follow-up analyses, using logistic regression, found that in examining all demographic, mental health and severity of substance use, the only factor that significantly contributed to early discharge was that participants who did not finish school beyond Grade 10 were more likely to discharge early (70%) than remain in treatment (30%; *p* = 0.021). 

The average number of Grit sessions attended in the first six week cycle was 5.93 (SD = 3.59; range = 0 to 12 sessions; 52.5% (*n* = 62) completed at least six sessions, 7.6% (*n* = 9) completed to the 12th session) and for the 3-month follow-up was 7.19 (SD = 5.29; range = 0 to 25 sessions; 57.6% (*n* = 68) completed at least six sessions and 18.6% (*n* = 22) completed to the 12th session. On average, participants attended 76% of Grit sessions available to them while in residential treatment. Main reasons for non-attendance included being exempted by staff (31.4%), being off property (18.8%), or having another appointment (16.9%). The main reason for not completing all 12 Grit sessions was exiting residential treatment early. 

### 3.3. Key Primary Outcomes

#### Substance Use

Analysis of the Global ASSIST score using mixed level modelling revealed an effect of time F(3, 301.98) = 64.37, *p* < 0.001, which was not qualified by a time by group interaction, F(3, 301.91) = 1.56, *p* = 0.199. Across both groups, there was a decrease in Global ASSIST score from baseline to the 3-month (MD = 44.46, 95% CI [35.49, 53.43], d = 0.80), 6-month (MD = 57.39, 95% CI [47.22, 67.56], d = 1.03), and 12-month follow-up (MD =49.87, 95% CI [−18.26, 3.22], d = 0.90). At baseline, the Grit group scored higher on the Global ASSIST score compared to the control group (MD = 19.91, 95% CI [3.48, 36.37], *p* = 0.018), however this difference is not significant when controlling for multiple comparisons.

Mixed level analysis of the Alcohol ASSIST Score also revealed an effect of time F(3, 361.95) = 24.86, *p* < 0.001, not qualified by a time by group interaction, F(3, 361.85) = 0.59, *p* = 0.622. Across groups there was a decrease in Alcohol score between baseline and 3-month (MD = 8.09, 95% CI [5.97, 10.22], d = 0.65), 6-month (MD = 7.61, 95% CI [5.39, 9.83], d = 0.61) and 12-month follow-up (MD = 5.73, 95% CI [3.52, 7.93], d = 0.46). 

For Methamphetamine ASSIST Score, there was an effect of time F(3, 237.36) = 41.08, *p* < 0.001, qualified by a significant time by group interaction, F(3, 237.18) = 2.90, *p* = 0.036. When examining the simple effects of time within each location, both groups had a significant effect of time, with F(3, 191.50) = 40.67, *p* < 0.001 for Grit and F(3, 200.40) = 9.90, *p* < 0.001 for the control group. The Grit group had a larger reduction in methamphetamine score from baseline to 3-months (see Table 3, MD = 12.62, 95% CI [10.22, 15.03], d = 0.96) than the control group (MD = 6.96, 95% CI [3.94, 9.97], d = 0.53). There was also significantly reduced Methamphetamine ASSIST Scores from baseline to 6-month and 12-month follow-ups for both Grit (at 6-month MD = 12.75, 95% CI [9.71, 15.79], d = 0.97 and at 12-month MD = 10.69, 95% CI [7.57, 13.81], d = 0.81) and Control group (MD = 7.97, 95% CI [4.26, 11.68], d = 0.60 and MD = 8.23, 95% CI [4.38, 12.07], d = 0.62, for 6- and 12-month respectively). At baseline, the Grit group scored significantly higher on Methamphetamine score compared to the control group (MD = 5.32, 95% CI [1.41, 9.22], *p* = 0.008).

For the Cannabis ASSIST score there was an effect of time F(3, 330.99) = 14.00, *p* < 0.001, again qualified by a time by group interaction, F(3, 330.70) = 3.38, *p* = 0.019. When examining the simple effects of time within each group, there was a significant effect of time for the Grit group (F(3, 286.05) = 18.56, *p* = < 0.001), but there was no overall effect of time for the control group (F(3, 300.67) = 2.11, *p* = 0.099). The Grit group reported a moderate reduction in cannabis from baseline to 3-months (see Table 3, MD = 7.30, 95% CI [5.09, 5.51], d = 0.62), 6-months (MD = 6.90, 95% CI [4.62, 9.18], d = 0.58), and 12-months (MD = 5.38, 95% CI [2.82, 7.94], d = 0.45). At baseline, the Grit group had a significantly higher Cannabis ASSIST score compared to the control group (MD = 4.95, 95% CI [1.44, 8.46], *p* = 0.006).

### 3.4. Other Outcomes

#### 3.4.1. Wellbeing

The linear mixed model revealed a significant effect of time, F(4, 449.29) = 17.82, *p* < 0.001, but no time by group interaction, F(4, 448.87) = 1.56, *p* = 0.183. Across groups, there was an increase in wellbeing from baseline to the 6-week (see Table 3, MD = 12.84, 95% CI [9.65, 16.03], *p* < 0.001, d = 0.73), and to the 3-month follow-up (primary outcome time-point; MD = 10.71, 95% CI [7.32, 14.10], d = 0.61). Wellbeing at the 6-month and 12-month follow-ups were also significantly higher than baseline (MD = 6.43, 95% CI [2.88, 9.99], *p* < 0.001, d = 0.37 and MD = 6.68, 95% CI [3.16, 10.20], *p* < 0.001, d = 0.38, respectively). 

#### 3.4.2. Depression and Anxiety

The linear mixed models revealed a significant effect of time for depression, F(4, 477.83) = 32.31, *p* < 0.001, and anxiety, F(4, 431.30) = 27.03, *p* < 0.001, but no time by group interaction for either depression, F(4, 477.71) = 1.03, *p* = 0.391, or anxiety, F(4, 431.20) = 1.84, *p* = 0.120. Across groups, there was a significant decrease in depression and anxiety from baseline to all follow-up (see Table 3, Mean difference was significant at *p* < 0.001 at all time-points). 

#### 3.4.3. Vocational Engagement

Mixed binary logistic regression analysis revealed a significant effect of time, F(3, 545) = 7.25, *p* < 0.001, not qualified by a time by group interaction, F(3, 545) = 1.23, *p* = 0.299. Across groups vocational engagement increased, from 26% at baseline to 33% at the 3-month follow-up, 41% at 6-month follow-up, and 46% at 12-month follow-up.

## 4. Discussion

The present study reports the outcomes of the novel strength-based Grit Wellbeing and Self-regulation Program delivered in a residential AOD treatment setting. Participants who received the Grit program in addition to standard residential care achieved significant decreases in global and specific substance use scores. There were no group differences in reductions for global substance involvement or alcohol involvement, but participants in receipt of Grit reported significantly greater reductions in cannabis involvement and methamphetamine involvement score than the control group; partially supporting the substance use hypothesis. The wellbeing, mental health, and vocational engagement hypotheses were not supported, with both groups showing similar improvements in wellbeing, depression, anxiety, and vocational engagement across time. Across all groups there was a marked reduction in substance use, depression and anxiety and an increase in wellbeing and vocational engagement. Further, this reduction was maintained 12 months post baseline. 

Considering the large treatment dose that clients already receive through residential treatment, it is impressive that Grit produced stronger treatment effects for cannabis use and methamphetamines. It is unclear why the intervention affected these specific substances. It is possible that the strategies offered in Grit—focusing on emotion regulation, mindfulness, social connection and strength-based practices—may have been particularly beneficial for people with these substances of concern. However, further mediation analyses are needed to determine the exact mechanisms that may have facilitated these changes. It is also possible that the higher methamphetamine and cannabis scores found in the intervention over control group may partially explain the finding, as the former had a greater potential for change due to the higher initial levels of use (regression to the mean). Nevertheless, the current findings suggest that further investigation into the benefits of Grit are warranted. With methamphetamine use being a prominent concern, particularly within Australia [58], the prevalence of cannabis use increasing internationally [59], and these drugs being the first and third most common presenting drugs of concern for clients entering residential treatment in this study, it would be beneficial to determine the efficacy of the Grit program with broader samples (adults and older adults, as well as youth) and across other settings (e.g., outpatient day programs) using more rigorous designs (e.g., randomized control trials).

Treatment attrition was a problem, as discharge prior to completion of 6-week residential treatment program was relatively common. The high discharge rates, which were found in both groups, is not unusual in substance use treatment [3]. Follow-up analyses found that fewer years of education was associated with attrition rates, which is consistent with some previous research [60,61]. However, other factors such as age, more severe substance use (i.e., higher scores for global substance use, alcohol, cannabis or methamphetamine use) or more serious mental health concerns (i.e., screening positive for PTSD or psychosis or higher scores for anxiety or depression) were not associated with early discharge, which counters some previous findings [62]. Retention in residential treatment is an ongoing challenge in recovery from substance use problems, and so is critical to address given strong evidence for better substance use outcomes with treatment retention and completion [63,64,65,66,67]. 

Despite the high attrition rates, there were still dramatic reductions in substance use across groups, with average reductions of at least 40% in global and specific substance use scores across groups post-treatment, and these reductions were maintained at 6- and 12-months. These findings highlight the positive outcomes that residential treatment can achieve and maintain, even in the face of early discharge. 

The results of the study need to be considered in light of a number of limitations. First, the inability to randomize participants to the different treatment facilities may have resulted in sampling biases. Although few differences were found between the groups on baseline characteristics and those found controlled in analysis, the baseline differences in cannabis and methamphetamine use may have been reduced if random sampling were employed. 

Another confound was potential differences between the two residential treatment facilities. While both facilities were operated by the same organisation, used the same six-week program treatment model, and engaged staff in similar training activities, there were differences between the facilities, in personnel, location, layout, and entry criteria into the facility (the control facility allowed all adults, while the intervention only catered for 18–35 year olds); all of which were difficult to control and may have impacted upon treatment outcomes. Additionally, due to the way the services operated, it was not possible to implement a randomized control trial to randomize clients to one service site or the other. As such, we were unable to control for confounding effects. For example, socioeconomic factors associated with site location, or potential other reasons for people requesting entry one service or the other, which lead to systematic differences between the sites in participant selection. Whilst randomization was not possible, other designs that may be utilized in future is time-based design using a single site or a cross-over design (although this may also have unintended contamination due to staff training in the intervention program). This was not feasible to conduct in the current study, due to timing restrictions of the study. Further, the fidelity of the usual six-week program was not assessed and details of treatment received as part of this program were not recorded. However, other factors that may have impacted on treatment outcomes, including length of stay, program completions, reasons for discharge and number of re-admissions, were examined across the two groups. The only difference found was in the number of re-admissions, which was controlled in all analyses. Another significant difference and potential confound between the sites was the read-mission rates as 3-months, which were significantly higher at the treatment site. We are unable to determine if this difference was a function of the intervention itself or other site-specific factors. Both facilities had the same organizational rules regarding readmission, but it is unknown how differences in staff and service factors may impact readmissions rates. These factors may include how discharge and admission processes are implemented by site staff, hospitality and welcomeness of the service, discharge locations, and convenience of location.

A fourth limitation was the attrition rate, with 74% of participants retained at the 3-month follow-up, 64% at the 6-month and 12-month follow-up. However, this is lower than the 30% attrition forecast at the primary time-point (3-months) and was comparable across groups. Nevertheless, while there was no difference between those who did not complete more than one survey and those who did, the reduced sample size at follow-up may have resulted in more favourable outcome reporting. 

Despite these limitations, this study had some notable strengths. First, it was conducted within the residential AOD treatment facilities, with a comprehensive training program for AOD workers. This implementation framework provided the opportunity to conduct translational research, and examine feasibility and implementation processes in a real world clinical setting. Second, we limited exclusion criteria to be inclusive, to increase the number of clients offered the intervention and to maximise external validity of the findings. Our sample comprised over 80% of all the young people who entered residential treatment during the study, and is likely to be both representative and reflective of the complexity of presentations of this client group. Including this large sample also presented an opportunity to better understand the complex comorbidities (social, mental health, educational) that commonly co-occur for this population. 

Given the findings of the current study, future research examining of the moderators of treatment effects and process outcome variables is warranted. In particular, it may be worthwhile to examine if demographic factors such as age and gender and individual factors like primary problematic drug selects, polysubstance use, or comorbidities upon entry the service, influenced the treatment effects. Further, given the Grit intervention was specifically targeting factors such as impulsivity and self-regulation, further research should be done to examine if these process outcomes were mechanisms of change. That is, to understand when and if the key components of Grit were a) successfully targeted in the program and b) related to primary outcomes, it is critical that future research examines if the intervention led to changes in proposed treatment targets such as emotion regulation, impulsivity, social support, and self-perceived strengths and how these factors predict substance use and wellbeing outcomes.

## 5. Conclusions

Substance use disorders are a significant cause of disability among young adults and can have severe adverse health outcomes [59,68]. Residential AOD facilities are a common treatment pathway to address moderate to severe substance use among help-seeking individuals [1,69], but there is a lack of evidence on best practice approaches to treatment in these settings [1,3,67]. Grit was developed with the aim of addressing this gap to better target the complex health and wellbeing needs of young people experiencing SUDs in residential AOD treatment settings. This study presents promising evidence that a six-week residential program can achieve beneficial reductions in AOD use, depression and anxiety and improvements in wellbeing and vocational engagement. Further, the Grit Wellbeing and Self-regulation Program may enhance the outcomes of such programs; particularly in relation to methamphetamine and cannabis use. 

## Figures and Tables

**Figure 1 ijerph-19-13835-f001:**
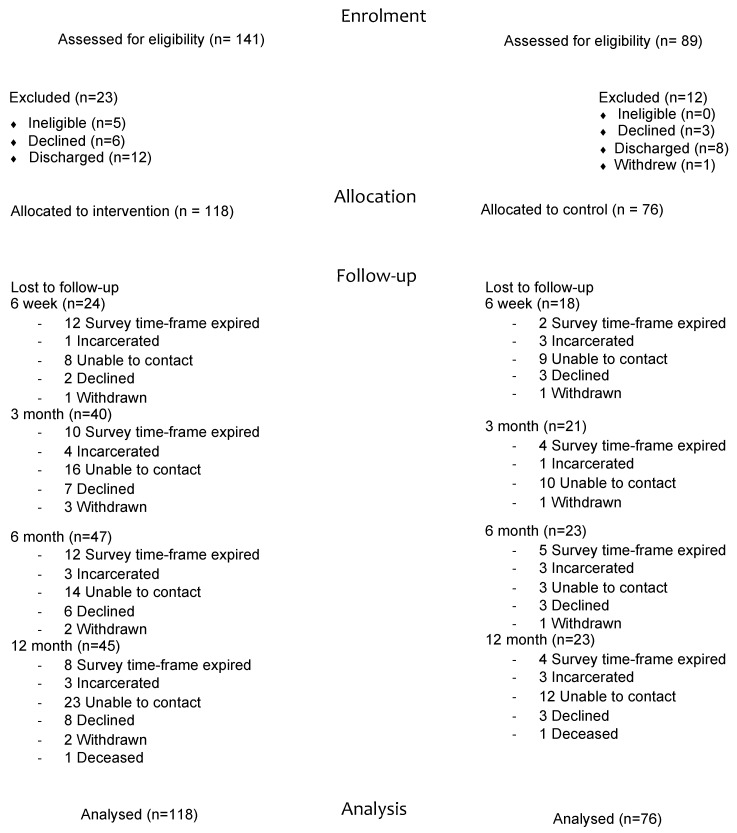
CONSORT.

**Table 1 ijerph-19-13835-t001:** Baseline sample characteristics.

Demographics ^1^	Total (*n* = 194)	Intervention (*n* = 118)	Control (*n* = 76)	*p*-Value
Age	27.40 (5.73)	26.60 (6.04)	28.61 (4.97)	0.021
Male	128 (66.0)	81 (68.6)	47 (61.8)	0.330
Australian born	193 (90.2)	108 (91.5)	67 (89.3)	0.610
Indigenous	21 (10.8)	13 (11.0)	8 (10.5)	0.915
Relationship—Single	152 (78.4)	92 (78.0)	60 (78.9)	0.871
Grade 10 schooling	106 (54.6)	67 (56.8)	39 (51.3)	0.417
Unemployed	143 (73.7)	85 (72.0)	58 (76.3)	0.509
Receiving Pension	159 (82.0)	92 (78.0)	67 (88.2)	0.076
Live with parents/relatives	82 (42.3)	51 (43.2)	31 (40.8)	0.738
Homeless	43 (22.2)	31 (26.3)	12 (15.8)	0.084
Have at least one child	67 (34.5)	32 (27.1)	35 (46.1)	0.007
Primary Drug				0.754 ^a^
Methamphetamine	87 (44.8)	53 (44.9)	34 (44.7)	
Alcohol	67 (34.5)	39 (33.1)	28 (36.8)	
Cannabis	29 (14.9)	19 (16.1)	10 (13.2)	
Other	11 (5.7)	7 (5.9)	4 (5.3)	
Mental Health				
PTSD Screen ^b^	84 (43.3)	52 (44.1)	32 (42.1)	0.834
Psychosis Screen ^b^	85 (43.8)	55 (46.6)	30 (39.5)	0.381
Depression ^c^	85 (43.8)	55 (46.6)	30 (49.5)	0.279
Anxiety ^c^	103 (53.1)	69 (58.5)	34 (44.7)	0.076
Correctional Involvement ^d^	41 (21.1)	24 (20.3)	17 (22.4)	0.722
Hospitalized ^d^	61 (31.4)	34 (28.8)	27 (35.5)	0.366

^1^ For categorical variables, frequencies are presented, followed by the percentage in parentheses. For the continuous variable of age the mean is presented with the standard deviation in parentheses. ^a^ difference in primary drug selected; ^b^ Positive screen; ^c^ Categorised as experiencing moderately severe symptoms; ^d^ past month.

**Table 2 ijerph-19-13835-t002:** Attendance at Residential Rehabilitation Services.

	Total *n* (%)	Intervention *n* (%)	Control *n* (%)	*p*-Value
**Early Discharge** ^a^	(*n* = 119)	(*n* = 74)	(*n* = 45)	0.122
Voluntary	65 (54.60)	38 (51.40)	27 (60.00)	
Involuntary	52 (43.70)	34 (45.90)	18 (40.00)	
Other	2 (1.60)	2 (2.70)	0 (0.00)	
**6 weeks**				
Completed 3 weeks ^a^	138 (71.70)	79 (66.90)	59 (77.60)	0.111
Completed 6 weeks ^a^	75 (38.70)	44 (37.30)	31 (40.80)	0.625
Average number of days- Residential ^b^	29.05 (13.66)	28.29 (13.84)	30.24 (13.38)	0.332
**3 months**				
Completed 3 weeks ^a^	143 (73.70)	84 (71.2)	59 (77.6)	0.321
Completed 6 weeks ^a^	95 (49.50)	59 (50.0)	37 (48.7)	0.858
Average number of days—Residential ^b^	37.71 (22.58)	37.25 (22.87)	38.41 (22.25)	0.728
Re-admissions ^a^	68 (35.0)	54 (45.8)	14 (18.4)	0.001
New Admission ^a^ Episodes	14 (7.22)	7 (5.93)	7 (9.21)	0.389
**12 months**				
Average number of days—Residential ^c^	43.90 (31.89)	46.22 (35.76)	40.30 (24.50)	0.598
New Admission Episodes ^a^	44 (22.68)	12 (15.79)	32 (27.12)	0.066

^a^ Categorical variable, with frequencies followed by the percentage in parentheses, *p*-values based on chi-square test of independence; ^b^ Continuous variable, the mean is presented with the standard deviation in parentheses, *p*-value based on *t*-test; ^c^ Continuous variable, the mean presented with the standard deviation in parentheses, values are non-normally distributed, *p*-value based on Mann–Whitney test.

**Table 3 ijerph-19-13835-t003:** Outcome Measures at Baseline and at 6 week, 3-, 6-, 12-month follow-ups.

		Baseline	6 Week	3 Month	6 Month	12 Month
		*M* (*SE*)	*M* (*SE*)	*M* (*SE*)	*M* (*SE*)	*M* (*SE*)
Global ASSIST	Grit	130.48 (5.39)	-	76.86 (5.98)	64.89 (6.52)	77.17 (6.40)
	Control	110.57 (6.47)	-	75.27 (7.22)	61.38 (7.81)	64.14 (7.65)
	Total	120.52 (4.24)	-	76.06 (4.70)	63.13 (5.09)	70.65 (5.00)
Total ASSIST	Grit	108.88 (4.93)	-	55.91 (5.50)	41.10 (6.02)	53.07 (5.94)
	Control	90.54 (5.93)	-	56.34 (6.66)	39.20 (7.16)	41.96 (7.05)
	Total	99.71 (3.88)	-	56.12 (4.32)	40.15 (4.68)	47.51 (4.62)
Alcohol ASSIST	Grit	17.49 (1.21)	-	9.31 (1.35)	11.12 (1.46)	12.5 (1.46)
	Control	18.66 (1.46)	-	10.65 (1.63)	9.8 (1.72)	12.19 (1.72)
	Total	18.07 (0.95)	-	9.98 (1.06)	10.46 (1.13)	12.35 (1.13)
Methamphetamine ASSIST	Grit	22.66 (1.28)	-	10.04 (1.40)	9.92 (1.51)	11.98 (1.54)
	Control	17.35 (1.54)	-	10.39 (1.69)	9.38 (1.79)	9.12 (1.84)
	Total	20.01 (1.01)	-	10.22 (1.10)	9.65 (1.17)	10.55 (1.20)
Cannabis ASSIST	Grit	15.54 (1.15)	-	8.24 (1.26)	8.64 (1.30)	10.16 (1.34)
	Control	10.59 (1.38)	-	8.48 (1.51)	7.55 (1.54)	7.37 (1.60)
	Total	13.07 (0.91)	-	8.36 (0.99)	8.10 (1.02)	8.76 (1.05)
Wellbeing (MHC-SF)	Grit	33.52 (1.69)	45.84 (1.82)	46.94 (1.93)	42.75 (2.17)	42.59 (2.07)
	Control	36.86 (2.06)	50.22 (2.28)	44.86 (2.36)	40.50 (2.47)	41.15 (2.41)
	Total	35.19 (1.34)	48.03 (1.46)	45.90 (1.52)	41.62 (1.65)	41.87 (1.59)
Depression (PHQ-9)	Grit	14.76 (0.73)	9.13 (0.77)	8.47 (0.79)	8.60 (0.85)	8.90 (0.85)
	Control	12.32 (0.87)	7.32 (0.95)	7.36 (0.98)	8.42 (0.98)	8.45 (1.00)
	Total	13.54 (0.57)	8.23 (0.61)	7.91 (0.63)	8.51 (0.65)	8.67 (0.66)
Anxiety (GAD-7)	Grit	12.46 (6.34)	8.63 (6.55)	6.66 (6.00)	6.40 (6.36)	6.35 (5.79)
	Control	10.18 (6.51)	6.25 (5.54)	6.56 (6.58)	7.36 (5.88)	7.74 (6.37)
	Total	11.58 (6.48)	7.76 (6.26)	6.62 (6.21)	6.81 (6.14)	6.90 (6.03)

Note: The ASSIST is a three month measure so was not assessed at 6 weeks. Vocational engagement is not included in this table due to being a binary variable. ASSIST: World Health Organisation Alcohol, Smoking and Substance Involvement Screening Test [49]: MHC-SF: Mental Health Continuum Short-Form [23]; PHQ-9: Patient Health Questionnaire 9-item [53]; GAD-7: Generalised Anxiety Disorder 7-item Scale [54].

## Data Availability

The data presented in this study are available on request from the corresponding author. The data are not publicly available due to privacy concerns and in accordance with the Ethics Approval of the study.

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
