# Peer review of "Controlled Trial Examining the Strength-Based Grit Wellbeing and Self-Regulation Program for Young People in Residential Settings for Substance Use"

_ijerph, 2022, doi:10.3390/ijerph192113835_

Round 1

Reviewer 1 Report

Controlled trial examining the strength-based Grit Wellbeing and Self-regulation Program for young people in residential settings for substance use

This research paper evaluates the Grit program, which aims to increase mental health by “building self-regulation skills, strengths, social connections, and health behaviours” by examining measures of substance use, mental health, and vocational achievement after receiving either standard care alone, or standard care in addition to the Grit program.

The authors review the mixed evidence for Mindfulness Based Relapse Prevention, which is reasonable considering that it is likely the best direct comparison for the Grit program. The authors argue that the use a more integrated, holistic program like the Grit program – which builds core skills such as self-regulation, affect regulation, and impulsivity - is likely to produce enhanced results. Their primary outcome is the reduction of alcohol and drug use, with ancillary interest in wellbeing, mental health, and vocational engagement.  

One difficulty with the study is that the design required comparison of two sites, each offering one of the two conditions (intervention and control). The authors take pains to explain why randomization (of clients to the sites, or clients within the sites) was not feasible. The authors claim that a previous comparison of service file notes found comparable demographic characteristics at the two sites, but that data is not presented. That said, the difference in age ranges between the sites  effectively makes one a “young” program, while the other is more “for all ages”. For this reason, it was not altogether surprising to see that “Participants in the control group were more likely to be older and to have a child compared to the intervention group”. One wishes the authors could have instituted a time-based design (e.g., treatment as usual vs. treatment+Grit in perhaps using an ABAB design), although there may have been logistical issues which made this infeasible (they may wish to speak to this). All things considered, the design chosen may in fact have been the best compromise, as it allowed for a focus on the Grit skills and goals at the one site.   

The team’s assessment approach is comprehensive and the longitudinal design of the study, entailing data collection pre, post, and at 3, 6, and 9 months is ambitious, and the analysis appears appropriate. The authors should also note: in Figure 1, there are a number of spots where information is missing and the totals don’t add up (e.g., in the Experimental group, the number excluded from eligibility is 23, but only 11 are accounted for in diagram). I would also have liked to see some sub-analyses (e.g., males vs. females; married vs. single, etc.), although given the complex data analysis already reported, this may have been a bit overwhelming. Still, one is led to wonder whether some of the differences between the groups may have revealed themselves here (e.g., is it possible that females benefited more than males from the Grit training?)

One key finding I would have liked to see discussed further was the significantly higher number of readmissions in the Intervention group. Nearly half of the Intervention group were readmitted, while only 1 in 5 of the control group were. While the authors attempted to control for this statistically, it does not change the fact that these are quite starkly different. At the very least, there should be some discussion trying to account for this effect. It is possible it did not even have to do with the intervention at all, per se. For example, due to the study design discussed above and not being able to counterbalance the intervention, it is possible that one of the sites was quite simply “different” in some way. It may have been more or less hospitable, had some difference in terms of staffing, been more or less convenient, etc. For one thing, homelessness was marginally different between the two sites, as was likelihood of receiving a pension, leading one to wonder if they are different in terms of socioeconomic variables more generally. On the other hand, perhaps the difference in readmission rates did actually reflect something about the intervention. Again, the study design makes this difficult to tease apart one way or the other. But at the very least, this difference is non-trivial and warrants focused discussion (perhaps as part of the Limitations describing differences between the sites?).

The Discussion section is broad-ranging and well written. It summarized the significant findings, discussed alternate explanations, covered the representativeness of the study population, as well as differences between the sites and other limitations. I found this section well done, particularly considering the broad range of results being discussed. That said, it might have been useful to see a bit of discussion as to how or why the key components of Grit, like impulsivity and affect regulation, did not appear to make a difference. This is counter-intuitive at the least.

Ultimately, while the results were likely not as broad as the authors likely hoped, limited primarily to substance use and not the other mental health and quality of life variables, I found their arguments for why this study is important to be convincing. As the authors note, this study adds to the growing body of evidence that treatment for alcohol and other drug abuse can result in significant, lasting improvements in mental health and quality of life, and it will be of interest to see how Grit training helps reinforce those effects in the future.

Author Response

We would like to sincerely thank the reviewer for their time and effort in providing a thoughtful and helpful review. We have carefully read each of the reviewers comments, taken on board the feedback, implemented changes in the manuscript. 

Re: Controlled trial examining the strength-based Grit Wellbeing and Self-regulation Program for young people in residential settings for substance use

Reply to Reviewer 1 Report

We would like to sincerely thank the reviewer for their time and effort in providing a thoughtful and helpful review. We have carefully read each of the reviewers comments, taken on board the feedback, implemented changes in the manuscript, and have addressed these comments below.

One difficulty with the study is that the design required comparison of two sites, each offering one of the two conditions (intervention and control). The authors take pains to explain why randomization (of clients to the sites, or clients within the sites) was not feasible. The authors claim that a previous comparison of service file notes found comparable demographic characteristics at the two sites, but that data is not presented. That said, the difference in age ranges between the sites effectively makes one a “young” program, while the other is more “for all ages”. For this reason, it was not altogether surprising to see that “Participants in the control group were more likely to be older and to have a child compared to the intervention group”. One wishes the authors could have instituted a time-based design (e.g., treatment as usual vs. treatment+Grit in perhaps using an ABAB design), although there may have been logistical issues which made this infeasible (they may wish to speak to this). All things considered, the design chosen may in fact have been the best compromise, as it allowed for a focus on the Grit skills and goals at the one site.   

Thank you for your comment and for your understanding and articulation of the difficulties in conducting a trial in this environment. As the reviewer noted, we were unable to feasibly conduct the study in a way that would reduce some of the differences between sites. We have taken on board the reviewer’s comments and have included this in the discussion. Specifically, we have added the below paragraph to the Discussion (page 12):

                Additionally, due to the way the services operated, it was not possible to implement a randomized control trial to randomize clients to one service site or the other. As such, we were unable to control for confounding effects. For example, socioeconomic factors associated with site location, or potential other reasons for people requesting entry one service or the other, which lead to systematic differences between the sites in participant selection. Whilst randomization was not possible, other designs that may be utilized in future is time-based design using a single site or a cross-over design (although this may also have unintended contamination due to staff training in the intervention program). This was not feasible to conduct in the current study, due to timing restrictions of the study.

The team’s assessment approach is comprehensive and the longitudinal design of the study, entailing data collection pre, post, and at 3, 6, and 9 months is ambitious, and the analysis appears appropriate. The authors should also note: in Figure 1, there are a number of spots where information is missing and the totals don’t add up (e.g., in the Experimental group, the number excluded from eligibility is 23, but only 11 are accounted for in diagram).

Thank you for pointing out this error – when we had copied across the text box for the diagram, the box had cut off the last category (Discharged prior to enrolment in study, n = 12). This has been fixed now.

I would also have liked to see some sub-analyses (e.g., males vs. females; married vs. single, etc.), although given the complex data analysis already reported, this may have been a bit overwhelming. Still, one is led to wonder whether some of the differences between the groups may have revealed themselves here (e.g., is it possible that females benefited more than males from the Grit training?)

The reviewer makes an excellent point regarding the data analysis. As the primary analyses were preregistered, we have only reported here our preregistered data analyses, particularly as that is what we had the power to analyse and report. As the reviewer mentioned, the reported analyses were also complex and related to multiple outcomes, thus the decision was made not to report further exploratory analyses. We have added in the discussion the importance of future research that examined moderators and mechanisms of treatment (page 13):

Given the findings of the current study, future research examining of the moderators of treatment effects and process outcome variables is warranted. In particular, it may be worthwhile to examine if demographic factors such as age and gender and individual factors like primary problematic drug selects, polysubstance use, or comorbidities upon entry the service, influenced the treatment effects. Further, given the Grit intervention was specifically targeting factors such as impulsivity and self- regulation, further research should be done to examine if these process outcomes were mechanisms of change. That is, examining if the intervention lead to changes in emotion regulation, impulsivity, social support, and self-perceived strengths and how these factors predict substance use and wellbeing outcomes.

One key finding I would have liked to see discussed further was the significantly higher number of readmissions in the Intervention group. Nearly half of the Intervention group were readmitted, while only 1 in 5 of the control group were. While the authors attempted to control for this statistically, it does not change the fact that these are quite starkly different. At the very least, there should be some discussion trying to account for this effect. It is possible it did not even have to do with the intervention at all, per se. For example, due to the study design discussed above and not being able to counterbalance the intervention, it is possible that one of the sites was quite simply “different” in some way. It may have been more or less hospitable, had some difference in terms of staffing, been more or less convenient, etc. For one thing, homelessness was marginally different between the two sites, as was likelihood of receiving a pension, leading one to wonder if they are different in terms of socioeconomic variables more generally. On the other hand, perhaps the difference in readmission rates did actually reflect something about the intervention. Again, the study design makes this difficult to tease apart one way or the other. But at the very least, this difference is non-trivial and warrants focused discussion (perhaps as part of the Limitations describing differences between the sites?).

Thank you for bringing this oversight to our attention. We agree that it is important to highlight and discuss this significant difference in the discussion. As such, we have added to the following to the discussion on page 12:

Another significant difference and potential confound between the sites was the read-mission rates as 3-months, which were significantly higher at the treatment site. We are unable to determine if this difference was a function of the intervention itself or other site-specific factors. Both facilities had the same organizational rules regarding readmission, but it is unknown how differences in staff and service factors may impact readmissions rates. These factors may include how discharge and admission processes are implemented by site staff, hospitality and welcomeness of the service, discharge locations, and convenience of location.

The Discussion section is broad-ranging and well written. It summarized the significant findings, discussed alternate explanations, covered the representativeness of the study population, as well as differences between the sites and other limitations. I found this section well done, particularly considering the broad range of results being discussed. That said, it might have been useful to see a bit of discussion as to how or why the key components of Grit, like impulsivity and affect regulation, did not appear to make a difference. This is counter-intuitive at the least.

This comment from the reviewer highlights the importance of examining if the program successfully targeted the proposed key components of Grit. As pointed out by the reviewer, this was missing from our discussion, and we have added the following section to address this important point (on page 13):

Further, given the Grit intervention was specifically targeting factors such as impulsivity and self-regulation, further research should be done to examine if these process outcomes were mechanisms of change. That is, to understand when and if the key components of Grit were a) successfully targeted in the program and b) related to primary outcomes, it is critical that future research examines if the intervention led to changes in proposed treatment targets such as emotion regulation, impulsivity, social support, and self-perceived strengths and how these factors predict substance use and wellbeing outcomes.

Ultimately, while the results were likely not as broad as the authors likely hoped, limited primarily to substance use and not the other mental health and quality of life variables, I found their arguments for why this study is important to be convincing. As the authors note, this study adds to the growing body of evidence that treatment for alcohol and other drug abuse can result in significant, lasting improvements in mental health and quality of life, and it will be of interest to see how Grit training helps reinforce those effects in the future.

Thank you for these thoughtful comments towards the study and the broader research area.

Reviewer 2 Report

Thank you for the opportunity to review the paper entitled “Controlled Trial Examining the Strength-based Grit Wellbeing and Self-regulation Program for Young People in Residential Settings for Substance Use”. The manuscript was well-written so I only have a few comments for the authors to consider further improving the manuscript:

·      The manuscript would benefit if the authors could elaborate more about the inclusion of the four components of wellbeing: emotional, social, psychological and physical for AOD treatment. This is essential information which helps readers understand why the Program includes these elements which are the key to AOD treatment.

·      The authors need to include relevant theories or the results of existing literature which leads them to make the hypothesis (lines 70-74).

·      All arrows in the CONSORT figure are missing.

·      In line 362: The authors used “…strength-based Grit Wellbeing and Self-regulation Program…”. Could the authors explain why they used “strength-based”? Is strength-based the focus of the intervention?

·      The manuscript will benefit if more discussion on the findings is included as there is  limited discussion in the present form of the manuscript.

Author Response

We have carefully read each of the reviewer’s comments, taken on board the feedback, implemented changes in the manuscrip. 

Re: Controlled trial examining the strength-based Grit Wellbeing and Self-regulation Program for young people in residential settings for substance use

Reply to Reviewer 2 Report

We have carefully read each of the reviewer’s comments, taken on board the feedback, implemented changes in the manuscript, and have addressed these comments below.

  • The manuscript would benefit if the authors could elaborate more about the inclusion of the four components of wellbeing: emotional, social, psychological and physical for AOD treatment. This is essential information which helps readers understand why the Program includes these elements which are the key to AOD treatment.

Thank you for highlighting this important gap in the introduction. We have provided more information on the need for treatment addressing wellbeing, as well as the reason for examining the four aspects of wellbeing below (page 1).

Despite some studies reporting the positive effects of residential AOD treatment, quality evidence of its efficacy is limited; particularly in young people [1-2]. There is a clear need to continue refining AOD treatment in residential settings to improve long-term outcomes. While many residential treatment programs focus on AOD use and mental health problems to some extent, they often lack a broader focus on social, emotional, physical, and psychological wellbeing. A recent systematic review on the effectiveness of residential treatment services for individuals with substance use disorders highlighted that current evidence suggest best practice approaches is residential treatment that integrates mental health treatment takes a holistic approach to improving the overall wellbeing of the individual (beyond substance dependence). This approach is also congruent with the World Health Organisation’s definition of health, which notes that health is the state of physical, mental, emotional, and social wellbeing, rather than just an absence of disease [2].

  • The authors need to include relevant theories or the results of existing literature which leads them to make the hypothesis (lines 70-74).

We have provided further information on the literature around transdiagnostic theory and interventions, which provide the basis for our hypothesis on page 2.

There is growing evidence for transdiagnostic treatments that cut across diagnostic boundaries to target the risk and protective factors which underlie multiple mental health problems. Theoretically, transdiagnostic approaches suggest that there are common latent factors that underlie a range of mental health and behavioural presentations, including SUD [10, 11]. Transdiagnostic factors related to substance use include impulsivity, sensation seeking, coping, self-control, and emotion regulation skills [12-18]. Evidence-based treatments that have addressed these transdiagnostic factors include transdiagnostic CBT [10, 19-21], mindfulness-based [22] and emotion-regulation focused interventions [14] for depression, anxiety and/or SUDs. However, limited research has combined interventions targeting the common mechanisms underlying AOD problems and mental ill-health, as well as the multidimensional components of wellbeing [2].

  • All arrows in the CONSORT figure are missing.

Thank you for picking up this error, this has now been fixed.

  • In line 362: The authors used “…strength-based Grit Wellbeing and Self-regulation Program…”. Could the authors explain why they used “strength-based”? Is strength-based the focus of the intervention?

Yes, strength-based is a key focus of the intervention. This has been made clearer in the introduction, on page 3:

A feature of the program is that it takes a strength-based approach, moving the focus away from the deficit or pathologised approach to substance use and focuses on the strengths and resources of the clients. Adopting a strength-based approach is congruent with the program aims of holistically targeting an individual’s wellbeing, and there is emerging evidence that the utilisation of a strength-based approach is effective for supporting an individual’s substance use recovery [18, 19].

  • The manuscript will benefit if more discussion on the findings is included as there is  limited discussion in the present form of the manuscript.

Thank you for your helpful comments around the discussion. We have added sections expanding on the design, potential limitations, and future research to provide a more thorough discission section.

On page 13, line 464-474:

Additionally, due to the way the services operated, it was not possible to implement a randomized control trial to randomize clients to one service site or the other. As such, we were unable to control for confounding effects. For example, socioeconomic factors associated with site location, or potential other reasons for people requesting entry one service or the other, which lead to systematic differences between the sites in participant selection. Whilst randomization was not possible, other designs that may be utilized in future is time-based design using a single site or a cross-over design (although this may also have unintended contamination due to staff training in the intervention program). This was not feasible to conduct in the current study, due to timing restrictions of the study.

On page 13, lines 479 to 486:

Another significant difference and potential confound between the sites was the read-mission rates as 3-months, which were significantly higher at the treatment site. We are unable to determine if this difference was a function of the intervention itself or other site-specific factors. Both facilities had the same organizational rules regarding readmission, but it is unknown how differences in staff and service factors may impact readmissions rates. These factors may include how discharge and admission processes are implemented by site staff, hospitality and welcomeness of the service, discharge locations, and convenience of location.

On page 13, lines 504 -515:

Given the findings of the current study, future research examining of the moderators of treatment effects and process outcome variables is warranted. In particular, it may be worthwhile to examine if demographic factors such as age and gender and individual factors like primary problematic drug selects, polysubstance use, or comorbidities upon entry the service, influenced the treatment effects. Further, given the Grit intervention was specifically targeting factors such as impulsivity and self-regulation, further research should be done to examine if these process outcomes were mechanisms of change. That is, to understand when and if the key components of Grit were a) successfully targeted in the program and b) related to primary outcomes, it is critical that future research examines if the intervention led to changes in proposed treatment targets such as emotion regulation, impulsivity, social support, and self-perceived strengths and how these factors predict substance use and wellbeing outcomes.

Thank you for the opportunity to revise the manuscript in light of your comments. We believe these revisions have strengthened the study.